# Functional and Rheological Properties of *Vicia faba* L. Protein Isolates

**DOI:** 10.3390/biom11020178

**Published:** 2021-01-28

**Authors:** Daniel Żmudziński, Urszula Goik, Paweł Ptaszek

**Affiliations:** 1Department of Engineering and Machinery in Food Industry, Faculty of Food Technology, Agriculture University in Krakow, ul. Balicka 122, 30-149 Kraków, Poland; daniel.zmudzinski@urk.edu.pl (D.Ż.); urszula.goik@urk.edu.pl (U.G.); 2Department of Carbohydrates Technology, Faculty of Food Technology, Agriculture University in Krakow, ul. Balicka 122, 30-149 Kraków, Poland

**Keywords:** *Vicia faba* L. seeds, protein isolates, rheology, functional properties

## Abstract

A protein isolate (85.5%) was obtained from the *Vicia faba* L. seeds. The main protein fraction, typical for the seeds of this plant, was found to be most numerous: Legumin (35 kDa) and Vicilin (45 kDa). The hydrodynamic and surface properties of isolate aqueous solutions were studied with the help of dynamic light scattering, ζ-potential, and tensometry in a wide range of concentrations and pH conditions. Selected functional properties, like foaming and emulsifying abilities, were studied. An increase of water solubility was shown with a raising pH, as well as a water holding capacity (WHC). The protein isolate showed a tendency to decrease the surface tension of water solutions, with high hydrophobicity and a negative charge of the isolate enhancing the foaming and emulsifying properties. The analysis of the concentration and the pH influence on selected functional properties indicated alkaline conditions as favorable for good foaming and emulsifying properties of the isolate and affected on their rheological properties.

## 1. Introduction

In the era of depleting natural resources and the accompanying population growth, a significant tendency of an increase in the production of animal protein has been observed. This requires increasingly higher energy expenditure, as well as increasing the acreage of arable fields for animal feed production. The solution to this problem may be the use of proteins from other sources, such as from plants. Legumes are an important source of protein (more than 20 g of protein per 100 g), as well as carbohydrates, fats, and health-promoting ingredients, which has a positive effect on the circulatory system. They have also been known to counteract neurodegenerative diseases [1]. Furthermore, vegetable proteins are valuable for their functional properties, specifically from the technological properties of protein preparations. Those that result from their affinity for water have been found to be particularly important [2,3]. The interactions of proteins in a solution is of key importance for their use in food because homogeneous dispersion is necessary for the stabilization of foams and emulsions and for the formation of appropriate structures [4,5]. Solubility and water holding capacity (WHC) are important indicators of the quality of proteins as a functional food ingredient. They have a significant impact on the way the protein is used in the product due to the possibility of shaping the texture and sensory properties of the product. The use of protein is determined by emulsifying and foaming properties, the ability to form gels [6,7,8]. One of the important functional features of protein preparations resulting from their chemical structure are the foaming properties [5]. Foams are thermodynamically unstable systems and disintegrate with time which is a result of bubble coalescence, disproportionation, and drainage. As such, the susceptibility of foams to these phenomena may be a determinant of quality and durability [9,10]. The ability of proteins to form foams is closely related to the solubility of proteins and properties affecting the surface tension [5]. The second group of food systems that can be stabilized with proteins are food emulsions [11]. The ability to form an emulsion depends on the ability of proteins to adsorb at the interface, which is determined by the protein solubility and hydrophobicity [6]. The stability of the emulsion also depends on the molecular weight of the protein and polysaccharides used as stabilizers, their charge, concentration, ionic strength, pH, and temperature [11,12,13,14]. The rheological properties of protein-stabilized emulsions differ from Newtonian, showing differentiated behaviors [15,16,17].

*Vicia faba* L. is mainly used as livestock feed more often than human food [18,19] and as a soil nitrogen enhancer in the Mediterranean and Far East. Based on the size of the seeds, three varieties of *Vicia faba* L. are distinguished: var. minor, var. equina, and var. major [20,21]. Due to the protein content in seeds, which is slightly lower than in popular soybeans, *Vicia faba* L. arouses more and more interest among researchers [3,22,23]. In addition to a high concentration of protein and carbohydrates, broad fava beans are characterized by a high content of fiber, vitamins, and substances that lower the concentration of triglycerides and cholesterol in the blood [24,25,26]. The proteins presented in the seeds consist of two main fractions: albumin, rich in sulfur amino acids and lysine, along with globulins, consisting of convicilin, vicilin, and legumin [27,28]. The isoelectric point of the *Vicia faba* L. proteins is around pH 4.0 [29] where the solubility is the lowest, but with increasing the pH, the solubility constantly increases reaching a maximum at pH 8.0 [30]. It can be found in the literature selected information about the foaming properties of fava bean proteins can [2,5,31,32], as well as the use of fava bean proteins to shape the rheological properties of gels [33], foams [5], dispersion [15], or viscoelastic properties at the O/W interface [34]. The research results are available in the publications despite the fact that they are fragmentary. The results indicate a highly functional potential of practical applications for fava bean proteins. Examining from this conclusion, there was found to be a lack of information in the literature on the properties of the fava bean protein isolate and the possibility of using it to create dispersed systems.

For this reason, the aim of this study was to obtain the *Vicia faba* L. protein isolate, determine its protein composition, and study the basic hydrodynamic and surface properties of protein isolate aqueous solutions. The study also gathered information on selected functional properties of isolate, which included rheological properties of the emulsion stabilized by protein isolate.

## 2. Materials and Methods

### 2.1. Production of Protein Isolates

*Vicia faba* L. seeds were first ground and then mixed with water in a ratio of 1:9 and extracted in an alkaline medium (pH 10.5, 1 M NaOH) with the addition of 0.25% Na${}_{2}$SO${}_{3}$ for 1 h at 20 ${}^{{}^{\circ}}$C while being constantly stirred (2000 RPM). After this time, the extract was cooled to 4 ${}^{{}^{\circ}}$C and then separated from the sediment and the fat fraction by centrifugation at 3000× *g* for 10 min. The supernatant was acidified to pH 4.0 (1 M HCl) to precipitate the protein at the isoelectric point with constant agitation until the pH level was achieved. The precipitate was centrifuged at 3000× *g* for 10 min, washed, and centrifuged again. The purified sediment was transferred to water and normalized to pH 6.8, and then it was frozen and lyophilized. The isoelectric point was determined by acidifying (1 M HCl) the alkaline (1 M NaOH) supernatant with an additional level of 0.5 from pH 10.5 to 3.0. The isoelectric point was determined by acidifying the alkaline supernatant with 0.5 steps from pH 10.5 to 3.0. The lyophilized protein isolate was dissolved in a phosphate buffer at pH 5.0; 6.8; 8.0 and held for 0.5 h, slightly stirring for rehydration. Then, the samples were analyzed. Total protein (N·6.25) was determined by the Kjeldahl method. The isolate obtained contained 85.5% of proteins.

### 2.2. Hydrodynamic and Surface Properties

#### 2.2.1. Electrophoretic Research

Electrophoresis was performed on a polyacrylamide gel by SDS-PAGE in a reducing medium produced by 2-mercaptoethanol [35]. A Vertical Mini-Vertigel 2 electrophoresis apparatus (Apelex), cooperating with the PS 608 power supply (Apelex), was used. A portion of the lyophilized protein was dissolved in deionized water (1% solution) and then mixed in Eppendorf tubes with the reducing solution (0.125 M TrisCl, 4% SDS, 20% glycerol, 2% 2-mercaptoethanol, pH 6.8). The sample was placed in a water bath for 90 s, and, after cooling, was centrifuged at 15,000× *g*; next, it was injected onto a previously prepared bilayer polyacrylamide gel. The thickening layer had a concentration of 4% and the separating layer was 12.5%. The denaturation procedure was also used for the standard proteins. DC separations at 25 mA were performed for 80 min at a voltage of 100 to 260 V. The SDS6H2 and SDS7 protein kits (Sigma-Aldrich, St. Louis, MO, USA) were used as standards. The gels after staining in Coomassie Blue R-250 solution and scanned. Gel scans were analyzed using the GelAnalyzer 2010a software.

#### 2.2.2. ζ-Potential

The electrophoretic mobility ($\mu_{e}$) of proteins (1% solution) was determined by Zetasizer Nano ZS Malvern device. The results were obtained in aqueous solutions at pH 5.0, 6.8, and 8.0. A Phosphate buffer was used in each case. The zeta potential (ζ) of isolate of protein was calculated based on Smoluchowski-Henry equation:(1)$$\zeta =\frac{3\eta }{2\varepsilon F(\kappa a)}\mu_{e},$$
where: the F(κa) is a dimensionless function of the parameter κa, the symbol *a* corresponds to the radius of the particle (i.e., hydrodynamic radius $R_{h}$), ε is a dielectric constant, η is the viscosity of solvent, $\mu_{e}$ is an electrophoretic mobility, and κ is the Debye length.

#### 2.2.3. Hydrophobicity of Isolate Proteins

The surface hydrophobicity (*H*) was determined by the fluorimetric method [36] with 1-anilino-8-naphthalene sulfonate (ANS) as the fluorescent indicator. Several solutions of the given hydrolyzate were prepared with a protein concentration of 0.025 to 0.5 g·L${}^{-1}$ and the ANS solution (8.0 mmol·L${}^{-1}$) was added to them. Fluorescence intensity (FI) was measured on a Cary-Eclipse spectrofluorimeter (Varian) at 390 and 470 nm for excitation and emission, respectively. The tests were carried out on 1% isolate solutions (converted to protein) at different pH 5.0, 6.8, and 8.0.

#### 2.2.4. Dynamic Light Scattering (DLS)

The Dynamic light scattering measurement was performed on 1% solutions of broad fava bean protein isolate (calculated for protein) made in a phosphate buffer with pH 5.0, 6.8, and 8.0. The solutions were filtered using a 1 mm syringe filter. A set consisting of an ALV-CGS3 goniometer, an ALV-5000/EPP digital autocorrelator (ALV-Laser Vertriebsgesellschaft GmbH, Langen, Germany) and a laser with a wavelength of 532 nm and a power of 50 mW (JDSU) were used to characterize the hydrodynamic properties of broad bean protein isolate solutions. The determination of the autocorrelation function took place in the range of measurement angles from 30${}^{{}^{\circ}}$ to 130${}^{{}^{\circ}}$, at the temperature of 20 ${}^{{}^{\circ}}$C. Based on the measurement results, the diffusion coefficients and the hydrodynamic radius were determined using the Stokes-Einstein equation, Equation (Equation 2), by the CONTIN software [37,38]:(2)$$R_{h}=\frac{k_{b}\cdot T}{6\pi \cdot \eta \cdot D_{ef}},$$
where: $k_{b}$ is the Boltzman constant, *T* is the absolute temperature in Kelvin, and η is the viscosity of the liquid at temperature *T*.

#### 2.2.5. Measurement of Surface Tension and Determination of the Value of the Relaxation
Time

Measurement of the equilibrium surface tension was carried out using the Du Noüy ring detachment method. The tests were performed with the use of a Sinterface STA-1 tensiometer, and the measurements were made in conditions of constant air humidity and at a temperature of 20 ${}^{{}^{\circ}}$C. The tensiometer was calibrated at the air/water interface, under which conditions the value of the measured surface tension of deionized water (reference liquid) was 72.1 ± 0.1 mN·m${}^{-1}$. The surface tension was measured for aqueous solutions of fava bean protein with a concentration of 10${}^{-7}$–10${}^{-1}$% and 1–5% at the temperature of 20 ${}^{{}^{\circ}}$C. All solutions were prepared under different pH conditions: 5.0, 6.8, and 8.0. The tests were carried out for all combinations of concentration and pH, in triplicate, and measuring changes in surface tension over time. The results of the measurements were used to determine the value of the fava bean protein diffusion coefficients. Proteins, due to their amphiphilic nature, usually migrate towards the interface in order to adsorb on it. As indicated in the literature, a diffusion-controlled adsorption model is used to describe the dynamic surface tension behavior [39]. In line with the approach proposed by Reference [40], firstly, the normalized dynamic surface tension should be determined, which is defined as:(3)$$\sigma_{n}=\frac{\sigma_{t}-\sigma_{\infty }}{\sigma_{0}-\sigma_{\infty }},$$
where: $\sigma_{t}$, $\sigma_{\infty }$, and $\sigma_{0}$ are the surface tensions at time *t*, t→∞ (equilibrium), and t=0, respectively. The change in the normalized surface tension $\sigma_{n}$ over time can be expressed by the time series:(4)$$\sigma_{n}=\sum_{i=1}^{n}a_{i}\cdot e^{-\frac{t}{\tau_{i}}},$$
where: $a_{i}$ is the constant, and $\tau_{i}$ is the relaxation time. The relaxation time $\tau_{1}$ appearing in Equation (Equation 4) characterizes the dynamics of the system, especially the diffusion of macromolecules [41]. For the parameters estimation, the Marquardt-Levenberg minimization procedure was used. The target function was formulated as:(5)$$\chi^{2}=\sum_{i=1}^{N}{\left(\sigma_{i}-a_{1}\cdot e^{-\frac{t_{i}}{\tau_{1}}}\right)}^{2}\to min,$$
where $a_{1}$ can be interprated as a starting value of surface tension, and $\tau_{1}$ as a dominating relaxation time. *N* is the number of experimental points measured in time for an isolate aqueous solution at selected pH conditions.

### 2.3. Functional Properties Study

#### 2.3.1. Solubility and Water Holding Capacity (WHC)

The solubility was tested in parallel with the water holding capacity (WHC). For this purpose, 2 g of the dried sample were mixed with 25 mL of deionized water at 20 ${}^{{}^{\circ}}$C, and the entire mixture was stirred for 1 h. After that time, the suspension was centrifuged (1000× *g*, 10 min). Then, 5 mL of supernatant was collected, frozen, lyophilized, dried under vacuum (70 ${}^{{}^{\circ}}$C, 20 h) over phosphorus pentoxide, and then weighed. The rest of the supernatant was discarded, and the residue in the centrifuge tube was weighed. Solubility was expressed as the mass of soluble substances derived from 1 g of product, and WHC as the amount of water (in grams) retained by 1 g of product. The protein and WHC solubility curves as a function of pH for the hydrolysates produced under optimal conditions were determined in such a way that 4 g of the sample were mixed with 50 mL of distilled water and mixed for 15 min in the first cases. After this time, the suspension was adjusted to the desired pH in the range of 4.0 to 8.0 by the addition of HCl (3 mol·L${}^{-1}$) or NaOH (3 mol·L${}^{-1}$) and further shaken for 1 h at 30 ${}^{{}^{\circ}}$C with possible pH correction taking place. Then, the entirety was quantitatively transferred to a volumetric flask (100 mL) and made up to the mark with distilled water and then centrifuged (1000× *g*, 10 min). The protein in the clear supernatant was determined by the Kjeldahl method [42]. The insoluble residue in the centrifuge thimble was then weighed, and the WHC was calculated from the weight gain.

#### 2.3.2. Preparation of Foams and Testing Their Basic Properties

Foaming properties were analyzed at 20 ${}^{{}^{\circ}}$C by passing air at a velocity of $V_{f}$ = 5.55 mL·s${}^{-1}$ through 100 mL of an aqueous isolate solution with concentrations of 1%, 3%, 5%. The time *t* (s) required for the formation of 1000 mL of foam was measured, then the gas supply was shut off, and the unused liquid $V_{1}$ (mL) was immediately removed; after 5 min, the volume of the released liquid fraction $V_{2}$ (mL) was measured. All measurements were triplicated.

Based on this data, the following functional indicators were calculated: foaming capacity, foam overrun, and liquid drainage.

The foaming capacity (FC) was expressed as the ratio (%) of its volume to the amount of gas necessary for its production [43]:(6)$$FC=\frac{1000\;\mathrm{mL}-V_{1}}{V_{f}\cdot t}\cdot 100\%.$$

The foam overrun (FO) was presented as the quotient of the gas volume in the foam to the volume of the liquid with which it was formed [44]:(7)$$FO=\frac{900\;\mathrm{mL}}{100\;\mathrm{mL}-V_{1}}.$$

Foam stability was investigated after 5 min to evaluate the amount of liquid drainage that had accrued (LD${}_{5}$); this was the determined ratio of the amount of water phase separated from the foam after 5 min from forming to the volume of liquid that was in the foam at the time of its completion [9,10].
(8)$$LD_{5}=\frac{V_{2}}{100\;\mathrm{mL}-V_{1}}\cdot 100\%.$$

#### 2.3.3. Preparation of Model O/W Emulsions and Stability Testing

The emulsifying properties were determined by the turbidimetric method [45] using self-modification. Twenty milliliters of a phosphate buffer (0.1 mol·L${}^{-1}$) at pH 7.0 containing the dissolved product in an amount corresponding to 0.25, 0.5, 0.75, 1, 2, 3, 4, 5% protein was added with 5 mL grape seed oil from local market, and then the whole was homogenized at 30,000 RPM for 1 min using a CAT Unidrive X1000D homogenizer (Ingenieurbüro CAT, Ballrechten-Dottingen, Germany) equipped with a T10F blade. Immediately after homogenization and after 5 min, 50 μL of emulsion was taken, which was diluted with 10 mL of 1% SDS solution, and the turbidity of $A_{0}$ and $A_{5}$ was measured against the SDS solution used for dilution. Turbidance measurements were performed on a spectrophotometer in 1 cm long cuvettes at λ = 500 nm. All measurements were triplicated.

The emulsifying activity index (EAI) was calculated according to Reference [45]: (9)$$EAI=\frac{2T}{\phi c},$$
where $T=\frac{2.303A_{0}}{l}$ is the turbidity of the sample, ϕ volume fraction of dispersed phase, *c* is the concentration of protein isolate in the aqueous solution, and *l* is the pathlength of a cuvette. ESI can be expressed as the interface (cm${}^{2}$g${}^{-1}$) that could be obtained with the use of 1 g of protein isolate as a emulsifying agent [45].

The emulsion stability index (ESI) was calculated in (min) according to Reference [46] from the formula:(10)$$ESI=\frac{A_{0}}{A_{0}-A_{5}}\cdot 5\min$$

#### 2.3.4. Rheological Studies of Emulsions

Emulsions prepared according to the recipe given in point Section 2.3.4 were subjected to rheological tests. They were carried out using the RS-6000 rotational rheometer (ThermoFischer, Karlsruhe, Germany), geometry (2${}^{{}^{\circ}}$, d = 35 mm)-plate, with a gap of 0.110 mm at a temperature of 20 ${}^{{}^{\circ}}$C. The flow curves were determined over a shear rate range between 0.1 s${}^{-1}$ and 500 s${}^{-1}$. All measurements were triplicated. In the presented work, the Cross state equation was used to describe the rheological properties:(11)$$\eta_{app}=\eta_{\infty }+\frac{\eta_{0}-\eta_{\infty }}{1+{\left(K\cdot \dot{\gamma }\right)}^{m}},$$
where: $\eta_{0}$ is the apparent viscosity at zero shear rate, Pa·s, $\eta_{\infty }$-apparent viscosity for infinite shear rate, Pa·s, $\dot{\gamma }$-shear rate, $s^{-1}$, *m*-flow behavior index, and *K*-relaxation time, *s*. The estimation of the parameters of rheological models containing time constants allows for determining the most probable characteristic time of the tested material. Determining the changes in the values of the constants in the Cross equations as a function of concentration allows for the formation of a view on the structure of rheological phenomena occurring during shearing. The Marquardt-Levenberg minimization procedure was used for the parameters estimation. Target function was formulated as:(12)$$\chi^{2}=\sum_{i=1}^{N}{\left(\eta_{i}-\eta_{app}\right)}^{2}\to min,$$
where $\eta_{0}$, $\eta_{\infty }$, *K*, and *m* were calculated. *N* is the number of experimental points on flow curve $\eta_{i}$ for experimental shear rates ${\dot{\gamma }}_{i}$.

### 2.4. Statistical Analysis

A two-way variance analysis was carried out, examining the physicochemical parameters of protein solutions. Further analysis (post hoc) was carried out using the HSD-Tukey test. All calculations were carried out at the significance level p=0.05. Calculations were carried out using package R [47].

## 3. Results

### 3.1. Hydrodynamic and Surface Properties

The protein content of isolate was 85.5%. SDS-PAGE in a reducing medium was used to separate proteins according to their electrophoretic mobility as a function of the molecular weight of the polypeptides.

The gel shown in Figure 1 shows band with molecular weight ca. 35 kDa and in the range of 21–19 kDa, which corresponds to the fraction 11S legumin α chain and β chain, respectively. Moreover, between 55–45 kDa bands characteristic for the 7S vicilin protein fraction and its main subunits with the masses 55 kDa and 45 kDa were observed. Above 60 kDa, the presence of a few bands were found, the most numerous of which was represented by the fraction with a molecular weight of 64 kDa, which may correspond to the convicilin fraction. This is reflected in the studies of other researchers [48]. Moreover, the obtained results seem to be in line with those obtained by Reference [49]. As a result of electrophoretic separation, they obtained several types of 11S protein subunits with molecular weights of 36, 49, and 51 kDa, as well as three types of basic subunits of 19, 20.5, and 23 kDa. Similar observations were made by Reference [50] finding the presence of subunits with masses of 37, 20.1, 20.9, and 23.8 kDa. Reference [28] found the presence of three main subunits of proteins 11S legumin-like α chain and β chain (approximately 38–40 and approximately 23 kDa) and 7S Vicilin (46–55 kDa) and Convicilin (>60 kDa). The amount of the latter depended on the type of *Vicia faba* L. taken for testing. As shown in Table 1 the value of ζ-potential decreased in experimental range of pH to the highest charge of −34 mV at alkaline pH. In contrast, the changes in the value of effective diffusion coefficient $D_{ef}$ showed the nonlinear influence of pH conditions on the hydrodynamic properties of protein isolate in an aqueous solution. The highest value of $D_{ef}$ was observed at pH 5.0 and, and as a consequence, the lowest value of hydrodynamic radius. This result is in line with the ζ = −20.86 mV measured at this pH conditions. The highest value of hydrodynamic radius $R_{H}=$ 30 nm was detected for a protein isolate solution at pH 6.8 and decreased again at pH 8.0. This behavior can be explained in the light of hydrophobicity results. High surface hydrophobicity might be attributed to a high exposure of amino acid residues related to macromolecule expansion in solution. Hydrophobicity (*H*) value was the lowest at pH 5.0 and the highest at pH 6.8, and this can confirm results obtained with the help of the DLS method.

### 3.2. Functional Properties

#### 3.2.1. Solubility and WHC

The solubility (g of soluble protein per 1 g of solution) and water holding capacity (WHC, g of water per 1 g of product) are shown in Table 2. The increase in solubility can be observed, especially in the conditions of a higher pH, and a similar tendency was visible in the case of WHC, excepting the value at pH 5.0 (Table 1).

#### 3.2.2. Surface Tension

The dependence of surface tension σ as a function of isolate concentration *c* (Figure 2a) and time (Figure 2b) are presented for selected solutions of bean protein isolates. The experiment’s results show the typical behavior of the protein in the solution, demonstrating the ability to adsorb at the air/water interface. As can be seen in Figure 2a, the final values of the surface tension of the tested protein solutions decrease as a function of the concentration. In the case of dilute solutions, regardless of the pH, the surface tension assumes values close to that of water.

The σ-log
*c* behavior of broad bean protein isolate solutions at pH 5.0, 6.8, and 8.0 is comparable to that seen with many surfactants (Figure 2a). However, there is no sharp break point as is usually seen for surfactants in their critical micelle concentration. Similar behavior was observed by Reference [51] for the globular protein. The trend changes at 1% concentrations, possibly due to surface saturation. Surface tensions decrease with increasing concentration at all pH values, and, for pH 6.8 and pH 8.0, they are similar. With an increase in the concentration of the broad bean protein isolate in the solution at pH 5.0, the surface tension decreases at a concentration of $10^{-4}$ to 68 mN·m${}^{-1}$, reaching values close to the initial surface tension at a concentration of $10^{-6}$ for pH 6.8 and pH 8.0. The lowest value of the surface tension (44 mN·m${}^{-1}$) was obtained at a concentration of 4% and 5% for pH 6.8 and pH 8.0. The obtained surface tension results are similar to those obtained by References [4,5]. The reduction in surface tension was associated with an increased amount of soluble protein molecules present that could be adsorbed at the air-water interface. From the results shown in Figure 2b, it can be concluded that 1% protein solution at pH 5.0 has higher surface tension values compared to 1% solutions at pH 6.8 and 8.0. The surface tension significantly decreased with time for solutions towards a protein concentration of 1%, which indicates diffusion; below 1% of the protein solution, the surface tension changes slightly.

Due to their amphiphilic nature, proteins tend to migrate towards the interface, which is thermodynamically favorable [52]. Comparing the interactions at pH 6.8, pH 8.0, and pH 5.0 shows only slight differences in the diffusion coefficients (Table 2), which are slightly higher near the isoelectric point. Therefore, the main processes appear to be the same.

#### 3.2.3. Foaming Properties

The foaming properties of the broad bean protein isolate are shown in Figure 3a. The volume of gas introduced into the solution increases with increasing protein concentration, which is most pronounced for the acidic pH of the solution. At the lowest pH of 5.0, near the isoelectric point, the solubility of the protein is the lowest (Table 1), but the foaming capacity (FC) was largely influenced by the protein concentration in the solution. In the remaining pH ranges, the effect of protein concentration was noticeable for the FC parameter, but was not substantial. The foam overrun (FO) shown in Figure 3b details the structure of the foam. The smaller it is, the more desirable the foam becomes with greater amounts of fine-porous structure along with strong walls. The slight alkalization of the environment from pH 5.0 to 6.8 resulted in an almost double reduction of this parameter for the 3% and 5% protein solutions. The increase in solubility resulted in better migration of the proteins to the interface and improved the quality of air bubbles.

Useful properties of the foam can be used in food processing, when it is sufficiently durable. The LD${}_{5}$ coefficient, which is a measure of its stability, represents the amount of liquid phase released from the foam. The lower the value, the more durable the foam. The drainage measurement correlates very well with the actual foam quality [2]. Analyzing the data on Figure 3c, it can be seen that one of the smallest amounts of liquid drainage was released by foams obtained from an isolate solution of 3% at neutral and alkaline pH. The least amount of liquid was released by the foam at pH 5.0 at the concentration of 5% protein in the solution. However, taking into account the earlier indicators characterizing foams and the application possibilities in the food industry, the best quality foam was obtained at pH 6.8 with a 5% protein content in the solution. The ability to form foams should also be considered in the aspect of surface tension that occurs in protein solutions. It can be noticed that an increase in the protein concentration significantly lowers its value from 72 mN·m${}^{-1}$, regardless of the pH of the environment. Nevertheless, an increase in pH from 5.0 to 6.8 causes a significant reduction in the value of the surface tension to 49 mN·m${}^{-1}$, but further alkalization does not give such on effect (48 mN·m${}^{-1}$). On the other hand, alkaline solutions of protein, even at its low concentration, were characterized by a lower surface tension than comparable solutions at a lower pH. A lower value of this parameter has a procreative effect on foams forming (Figure 3a), but to a lesser extent on their stability. This is also related to the hydrodynamic parameters Table 1. The highest effective diffusion coefficient of proteins’ molecules in solutions was detected at pH 5.0. In these conditions, the protein chain was not expanded and its hydrophobic and hydrophilic zones were not available. The increase in pH caused an order of magnitude increase in the hydrodynamic radius and hydrophobicity with a simultaneous decrease in surface tension, which improves the ability of such solutions to form foams. The stability of the foams was the highest at the concentration of 1% of protein in the solution with a pH 6.8, due to the fact, that the hydrodynamic radius of the particles and the availability of the hydrophobic and hydrophilic zones were the highest (Table 1). Protein isolate solubility at pH 6.8 was much higher than at pH 5.0 (0.155 g·g${}^{-1}$), as well as WHC (3.469 g·g${}^{-1}$). Further alkalization caused a rising of effective diffusion coefficient value by the same hydrophobicity, which, in consequence, increased the volume of the foam and the degree of its gas filling, but it did have a smaller effect on the stability of the system.

#### 3.2.4. Emulsifying Properties

Emulsions O/W were prepared on the base of protein isolate aqueous solutions with selected concentrations at different pH levels. The influence of isolate concentration and pH on emulsifying activity index (EAI) and emulsion stability (ESI) are shown on Figure 4a,b, respectively. The increase in the protein concentration in each solution (regardless of its pH) increased the EAI emulsifying activity from 0.02 m${}^{2}\cdot$g${}^{-1}$ to 7.14 m${}^{2}\cdot$g${}^{-1}$ for pH 5.0, from 0.19 m${}^{2}\cdot$g${}^{-1}$ to 6.07 m${}^{2}\cdot$g${}^{-1}$ for pH 6.8 and from 0.26 m${}^{2}\cdot$g${}^{-1}$ to 9.14 m${}^{2}\cdot$g${}^{-1}$ for pH 8.0 achieving the highest value (Figure 4a). In addition, the alkalization of the environment favored the formation of the emulsion. For each of the concentrations, the EAI values were higher in the alkaline environment. Similar behavior was noted by Reference [7] for the kidney bean and field pea proteins. The main reason for high values of EAI is the solubility of the protein, significantly affected by pH, ζ-potential, and hydrophobicity of proteins (Table 1).

The increase in negatively charged chains with the simultaneous increase in hydrophobicity could be the result of the ionization of non-protein groups contained in the isolate, as well as from hydrophobicity originating from the development of the protein chain and exposure of its hydrophobic parts [5].

The disproportions in the emulsification stability (ESI) as a function of pH are noticeable from the concentration of 0.75% of the protein, with the highest stability recorded in the alkaline environment (Figure 4b). For a 1% solution of ESI protein at pH 5.0, it was 5.4 min at pH 6.8 10.3 min and at pH 8.0 119 min. However, the stability of the emulsion was variable as a function of the protein concentration. While the first three that obtained emulsions were not stable regardless of the pH of the solution (with the exception of 0.5% protein at pH 5.0 ESI 30 min), the range of protein concentration from 1 to 4% in the solution gave stable emulsions, especially in the case of alkaline solutions (ESI > 300 min). Of interest is the low stability of the emulsion obtained from a 5% solution, as well as a sharp increase in stability at 2% protein (ESI 65.5 min) and a slow decrease in ESI with increasing concentration of protein (5% ESI 6.4 min) for solutions with pH 6.8 and a slight increase in the stability of the emulsion in solutions with pH 5.0 (ESI 17.8 min 4% protein). The observed increase in emulsion stability as the protein isolate concentration decreased at pH 6.8 could be due to the greater degree of unfolding of the polypeptides in suspensions with a lower protein content during emulsification. This, in turn, made more protein surface available. In the case of soybean proteins, a similar relationship was observed [53]. The emulsions made in an alkaline environment with a 3% and 4% protein solution showed the greatest stability, achieving ESI values of 354 and 343 min, respectively [7]; the highest emulsion stability was obtained at pH 7.0 for kidney bean and field pea proteins, but the emulsion stability increased sharply from pH 5.0 (later dropping after exceeding pH 7.0), and further alkalization of the environment destabilized the emulsion. The maximum stability of the emulsion depended on the protein concentration and the pH of the solution. While at pH 5.0, where the amount of dissolved protein in the solution was the lowest, the max ESI was obtained at the highest 4% protein. On the other hand, the increase in pH to 6.8 resulted in the highest ESI at the protein concentration of 2%. Alkaline solutions gave more stable emulsions regardless of concentration, with a maximum of 3% protein.

### 3.3. Rheological Properties

Figure 5 shows the Cross model fit to the experimental data for emulsions with 1% and 5% protein isolate content for pH of 6.8 and for pH 8.0. The apparent viscosity of all emulsions tested decreased with increasing shear rate and showed non-Newtonian behavior. All emulsions behaved as shear thinning systems. The parameters obtained for the model are presented in Table 3. The rheological properties of the emulsion in the flow region differed depending on the pH of the aqueous phase used and the protein content in the dispersion phase. The influence of the pH of the continuous phase on the rheological properties is complex, and it results from the conformation of protein chains and the availability of hydrophobic and hydrophilic fragments of the macromolecule. In turn, the dependence of the rheological character of the emulsion on the protein isolate concentration seems to be less complex. It was found that the higher the isolate content in the aqueous phase, the higher the apparent viscosity of the emulsion was observed. The reasons for this phenomenon can be found in two facts. Firstly, a higher concentration of protein isolate increases the viscosity of the dispersion phase, according to the newtonian nature of protein isolate solutions. Secondly, the pH influence on apparent viscosity affected flow curves for emulsions stabilized by a 1% isolate solution. When comparing a 1% sample at pH 6.8 and pH 8.0 (Figure 5), higher values of apparent viscosity can be observed at lower pH environment. It can be implicated by the highest values of hydrodynamic radius and hydrophobicity (Table 1).

The higher viscosity of the continuous phase has a positive effect on the stabilization of the O/W emulsion. Moreover, the higher concentration of the fava bean protein isolate enhanced the stability of the interface in the oil-water system due to the high concentration of hydrophobic elements of the protein chains. The effects differ based on the concentration of the protein isolate with the rheological behavior and are reflected in the values of the parameters in the Cross rheological models (Table 3) during the continuous phase of pH 6.8, as well as 8.0. In the case of the Cross model, it can be noticed that the increase in the concentration of the isolate generates higher values of the viscosity $\eta_{0}$ and $\eta_{\infty }$. The values of the *K* time constant from the Cross model confirmed the increase in emulsion stability with the concentration of the isolate. The values of this constant can be interpreted as the relaxation time which, in the case of an emulsion, can indicate creaming and thus a tendency towards spontaneous delamination. In the case of pH 5.0, the values of $\eta_{0}$ and $\eta_{\infty }$ decrease with increasing concentration of the protein isolate. In the case of an emulsion made from a 5% isolate solution, the constant *K* has the highest value. The time constant (*K*) is related to emulsion instability (creaming). Thus, emulsions showing longer times showed greater stability due to strong drop-drop interactions, which can also be observed in Reference [17].

## 4. Discussion

*Vicia faba* L. protein isolate was good soluble in water in a wide range of pH. The values of WHC obtained confirmed the results of other researchers [54,55]. Some researchers [3] noticed greater water retention in broad bean flour than in the isolate itself, explaining it by the influence of a large amount of fiber. However, their isolate containing 92% of protein showed a lower water absorption capacity than that obtained by the authors. Perhaps the degree of protein purification (92% vs. 85.5%) could have made the difference. For *Pisum sativum* L. (vicilin) and legumin) proteins, a sharp increase in solubility was recorded from pH 7.0 [56]. The tested broad fava bean protein isolate before lyophilization was subjected to pH normalization at the level of 6.8 because it may contribute to the improvement of the functional properties of proteins, especially WHC [55]. The influence of pH on fava bean protein isolate properties can be explained in the light of hydrodynamic and surface results’ analysis. Alkaline conditions implicated high values of ζ and hydrophobicity, as well as hydrodynamic radius and low surface tension. The presented in Table 1 properties of the fava bean protein isolate as a function of pH showed changing hydrodynamic behavior. The hydrodynamic radius determined on the basis of dynamic light scattering for acidic conditions (pH 5.0) reached the lowest value of 3 nm. For conditions close to neutral (pH 6.8), the highest $R_{h}$ value of 30 nm was observed, which could be related to the aggregation of protein molecules [57]. A further increase in pH (8.0) caused aggregate dissociation and a decrease in the value of the hydrodynamic radius $R_{h}=$ 10 nm. The phenomenon is also reflected in the change in the diffusion coefficient ($D_{ef}$). This could mean that studied proteins could be more mobile in an acidic and alkaline environment. An increase in the pH value causes a sharp increase in the hydrophobicity (*H*) and a decrease in the value of the surface tension. As a consequence, high foaming capacity, low overrun at alkaline conditions were observed. With the increase of pH, an increase in FC (foaming capacity) was observed (the greatest of which was over 5 times for a 1% protein solution). Similar results were obtained by References [2,31]. They observed an increase in the foaming capacity as a function of pH and with an increase in protein concentration in the solution. Arogundade et al., in their research [30], noticed a greater stability of the *Vicia faba* L. protein isoalte foams at a pH close to the isoelectric point (IEP). In contrast, Reference [2] noticed a decrease in the amount of liquid drainage from the foam with an increase in protein concentration and with an increase in environmental alkalinity. The increase in foaming with the alkalization of the environment can be attributed to the increased flexibility of proteins, which may be due to charge interactions causing the repulsion of the molecules, the protein-protein system (Table 1). This, in turn, may result in better stabilization of the gas-liquid interface. The tested lyophilized protein isolate was not depleted of fat by extraction with organic solvents, but only by precipitation at low temperature. The presence of other isolate components can limit the effective diffusion coefficient of proteins and effected its adsorption on the interface [32].

The study on fava bean protein isolate properties showed its ability to create stable emulsions O/W type. The *Vicia faba* L. protein isolate consisted mainly of legumin α and β and a minor part of vicilin. The presence of vicilin itself could possibly contribute to better emulsifying properties [58]. Surface charge (ζ-potential), solubility, and hydrophobicity of isolate affected the emulsifying properties. The lower the values of any of these indicators, the worse the properties of the emulsion obtained will be [6]. Such a relationship was observed in the obtained O/W emulsions. The increase in environmental alkalinity caused an increase in negative charge and hydrophobicity, opening hydrophobic centers (increasing the hydrodynamic radius) and reducing surface tension. When comparing the hydrodynamic parameters of 1% aqueous solution of fava bean protein isolate (Table 1) with the values of Cross model parameters (Table 3) estimated for emulsion based on 1% of fava bean protein isolate concentration in dispersing phase, some tendencies can be visible. The low value of hydrophobicity and emulsifying activity index (EAI) at pH 5.0 (Figure 4a) could implicate larger diameter of dispersed oil phase, higher availability (in the meaning of effective concentration) of protein in water phase, and result in the highest $\eta_{0}$ and $\eta_{\infty }$ viscosities. The apparent viscosity of 1% emulsion at pH 6.8 were higher than at alkaline conditions. The hydrodynamic radius of protein isolate at pH 6.8 took the highest value of 30 nm and, with high interfacial surface (EAI = 1.0 m${}^{2}$ g${}^{-1}$), could influence on high zero shear rate viscosity due to strong friction effect at the beginning of flow. In turn, the properties of the same emulsion at pH 8.0 were shaped by higher interfacial surface (1.2 m${}^{2}$ g${}^{-1}$), which implicated smaller diameter values of oil phase droplets at the same concentration of all components. This fact in combination with smaller hydrodynamic radius of protein isolate determined lower apparent viscosities of 1% emulsion at alkaline condition in the range of higher values of shear rate. The influence of fava bean protein isolate concentration on increasing viscosity can be explained in the light of EAI values. The higher EAI value, the larger interfacial surface and the smaller diameter of oil phase droplets in emulsion. The larger interfacial surface implicates stronger friction effect during the flow, and, in consequence, higher values of viscosity are observed.

## 5. Conclusions

As shown by the test results, the fava bean protein isolate shows the ability to produce stable foams at a pH similar to the environment of many food products. In addition, such an environment promotes the stability of the emulsion at a low concentration of proteins in the solution. An increase in apparent emulsion viscosity was also observed, along with an increase in the protein isolate content in the aqueous phase. This phenomenon has a positive effect on the stabilization of O/W emulsions. Based on the research on functional properties, it can be concluded that the obtained isolates can be used in the production of foams and emulsions based on proteins of plant origin, both as an auxiliary factor and the main protein component of new products.

## Figures and Tables

**Figure 1 biomolecules-11-00178-f001:**
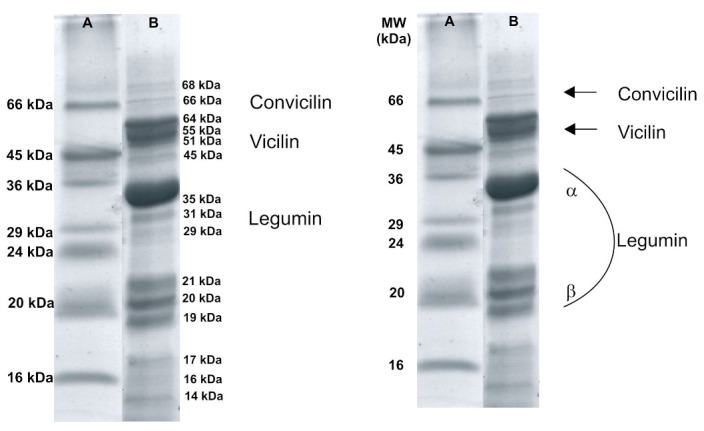
SDS-PAGE profiles of *Vicia faba* L. under reduction conditions. Lane: A-protein markers; B-*Vicia faba* L. protein isolate.

**Figure 2 biomolecules-11-00178-f002:**
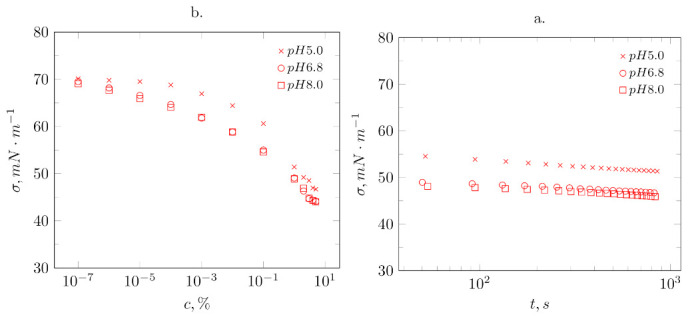
Surface properties of *Vicia faba* L. protein isolate solutions for various pH; (**a**) surface tension in function of concentration, (**b**) surface tension in function of time for 1% protein solution.

**Figure 3 biomolecules-11-00178-f003:**
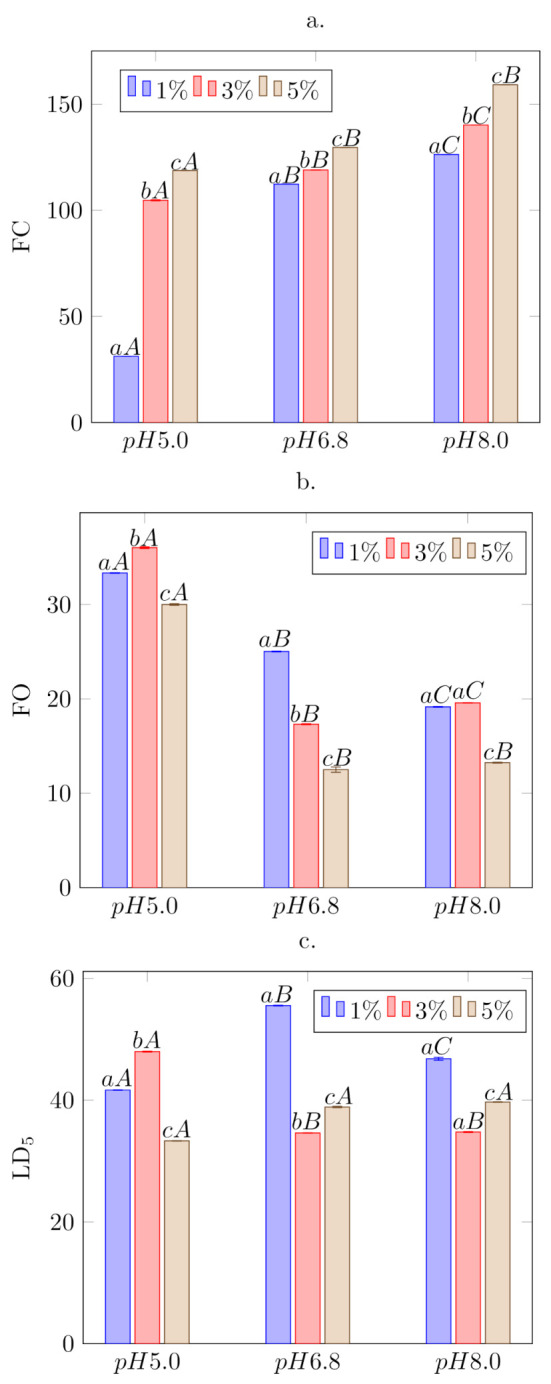
Foaming parameters of *Vicia faba* L. protein isolate solutions; (**a**) FC (foaming capacity), (**b**) FO (foam overrun), (**c**) LD${}_{5}$ (liquid drainage). Small letters represent analysis in one pH for different protein isolate concentrations, big letters represent analysis of pH influence for selected isolate concentration (1%, 3%, and 5%). Different letters indicate statistically significant differences between the groups (*p* < 0.05). Lowercase letters represent changes within one pH; capital letters represent changes of individual concentrations at the various pH.

**Figure 4 biomolecules-11-00178-f004:**
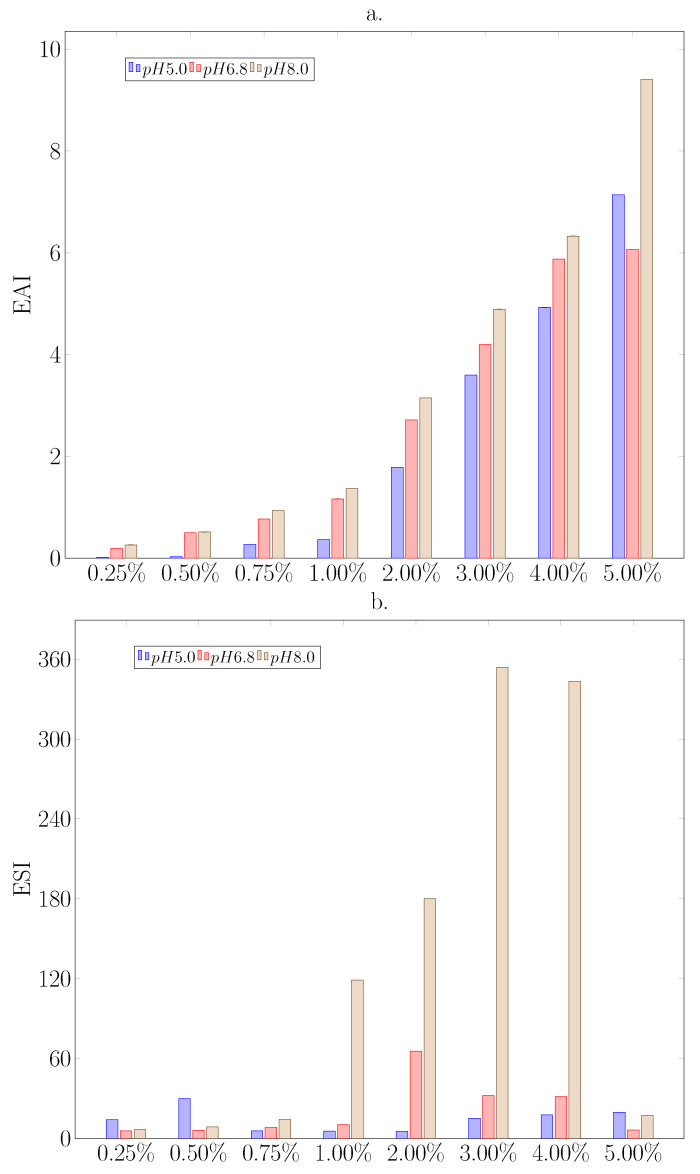
Emulsifying parameters of *Vicia faba* L. protein isolate solutions; (**a**) emulsion activity index (EAI), (**b**) emulsion stability index (ESI, min) as a function of isolate concentration and pH at 20 ${}^{{}^{\circ}}$C.

**Figure 5 biomolecules-11-00178-f005:**
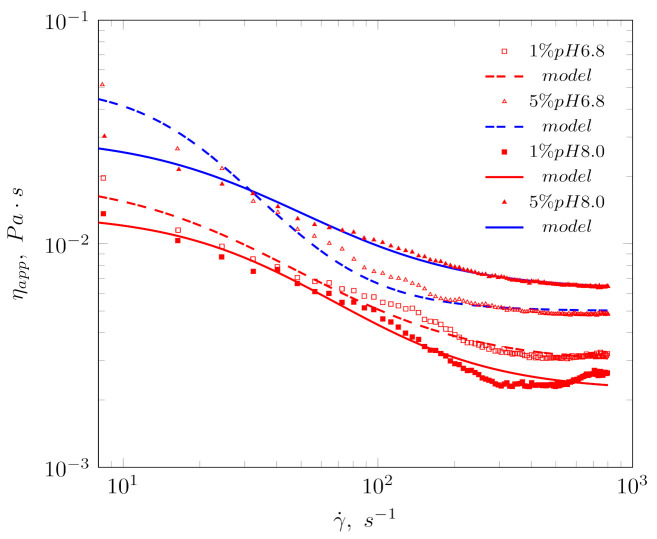
Flow curves and fitted Cross model Equation (Equation 11) of selected emulsions.

**Table 1 biomolecules-11-00178-t001:** Surface, hydrodynamics parameters, and hydrophobicity of *Vicia faba* L. protein isolate solutions.

		Surface Properties			Hydrodynamic Properties			
pH	c,%	$a_{1}$, mN·m${}^{-1}$	$\tau_{1}$, s	$\chi^{2}$	$D_{\mathrm{ef}}$, mm${}^{2}\cdot$s${}^{-1}$	$R_{h}$, nm	ζ, mV	H
5.0	0.001	72	186,000	0.01	-	-	-	-
	0.01	67	18,800	0.35	-	-	-	-
	0.1	64	14,800	1.25	-	-	-	-
	1.0	54	15,700	2.32	$7.80\times 10^{-9}$	3	−20.86	1.64
6.8	0.001	72	72,400	0.17	-	-	-	-
	0.01	66	18,000	0.61	-	-	-	-
	0.1	57	6800	1.19	-	-	-	-
	1.0	49	17,600	0.57	$0.80\times 10^{-9}$	30	−28.73	7.49
8.0	0.001	60	16700	0.75	-	-	-	-
	0.01	56	8200	1.17	-	-	-	-
	0.1	55	5700	2.45	-	-	-	-
	1.0	48	17,900	0.31	$1.50\times 10^{-9}$	10	−34.03	7.40

$a_{1}$, $\tau_{1}$ are the estimated parameters of Equation (4), $\chi^{2}$—least squares error sum of Equation (4),$D_{\mathrm{ef}}$—effective diffusion coefficient estimated with the help of CONTIN, $R_{h}$—hydrodynamic radius calculated according to Equation (2).

**Table 2 biomolecules-11-00178-t002:** Solubility and water holding capacity (WHC) in function pH for *Vicia faba* L. protein isolate.

			pH		
	4.0	5.0	6.0	6.8	8.0
Solubility	0.025 ± 0.003	0.040 ± 0.003	0.049 ± 0.003	0.155 ± 0.004	0.223 ± 0.001
WHC	2.255 ± 0.002	1.991 ± 0.001	2.585 ± 0.002	3.469 ± 0.002	3.494 ± 0.002

**Table 3 biomolecules-11-00178-t003:** The values of Cross model parameters.

pH	c,%	*K*, s	m, -	$\eta_{0}$, Pa·s	$\eta_{\infty }$, Pa·s	$\chi^{2}$
5.0	1	0.0003	2.85	1.384	0.014	0.09048
	3	0.0002	2.85	1.049	0.005	0.02977
	5	0.0051	1.68	0.124	0.005	0.00052
6.8	1	0.0107	1.45	0.020	0.003	0.00002
	3	0.0053	1.72	0.045	0.004	0.00007
	5	0.0032	1.94	0.052	0.005	0.00010
8.0	1	0.0022	1.73	0.014	0.003	0.00001
	3	0.0064	1.38	0.020	0.004	0.00001
	5	0.0075	1.47	0.030	0.006	0.00002

$\chi^{2}$—least squares error sum of Cross model fitting Equation (11).

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
