# Peer review of "Functional and Rheological Properties of Vicia faba L. Protein Isolates"

_biomolecules, 2021, doi:10.3390/biom11020178_

Round 1
Reviewer 1 Report
A review of the paper " Functional and rheological properties of Vicia faba L. protein isolates" by Daniel Zmudziński, Urszula Goik, and Paweł Ptaszek
I will give just a few examples of the drawbacks of this manuscript. Reading these results, I wonder if the authors have any basic knowledge of plant biochemistry. We read in manuscript legumin fraction (not “Legumina”) is 35 kDa and vicilin fraction (not “Vicilina”) is 45 kDa it is not precise. However, the weakest side of the manuscript is the discussion.
The authors studied changes in content of proteins. This is a very interesting and important problem. However, the authors describe their research very poorly. Description of the methods is imprecise. Actually, nothing is known. How were carried out alkaline medium or the supernatant was acidified? How were electrophoresis carried out? How were samples prepare for electrophoresis? Were samples cleaned? Such questions can ask the authors still very much. No statistical analysis of the results was conducted.
However, the gels on figure 1 are the weakest. This quality of gels should never be published.
Author Response
Thank you for review. We checked manuscript according to reviewer suggestions. We want to explain some problems with manuscript. Probably in LATEX template was mistake (we used template 13 Nov. 2020). In our best knowlage now every things is allright.
We expanded discussion and prepred new version of gel.
We think that now our manuscript represent better scietific level.
One more we are thankful for this review.
Reviewer 2 Report
The paper by Zmudzinski et al. presents a carefully-accomplished study to obtain the Vicia faba L. protein isolate, determine its protein composition and study the basic hydrodynamic and rheological properties of protein isolate aqueous solutions. In my opinion, the research is very well done, and the results are relevant to the biophysical community as they provide a needed benchmark for any study on the relationship between protein isolate structure and properties, including rheological properties.
Especially interesting from my point of view is the attempt of the authors to find a relationship between the structural features of the system with the rheological properties, mainly those related to the viscosity and the stability. Of course, this is a quite interesting approach that it should be explored in detail. I LIKE VERY MUCH THE VISTOSITY AT INFNITE SHEAR RATE….but I like much more the Newtonian viscosity. This is a physical feature related to the concentration, nature of dispersed phase of the solution, temperature….I expect a more deep study of the effect of both concentration and pH in rheological properties in future works.
Anyway, I believe that the findings represent a significant advance in the biomolecules and biophysics fields. Moreover, the overall quality and clarity of the manuscript id good, and the conclusions are supported by the presented results.
Author Response
Thank you for review. We checked manuscript according to reviewer suggestions. We want to explain some problems with manuscript. Probably in LATEX template was mistake (we used template 13 Nov. 2020). In our best knowlage now every things is allright.
We added discussion about meaning of eta_0 and eta_inf and influence on this parameters pH.
We think that now our manuscript represent better scietific level.
One more we are thankful for this review.
Round 2
Reviewer 1 Report
The manuscrypt should be published in this version